# Patient Education Improves Pain and Health-Related Quality of Life in Patients with Established Spinal Osteoporosis in Primary Care—A Pilot Study of Short- and Long-Term Effects

**DOI:** 10.3390/ijerph20064933

**Published:** 2023-03-10

**Authors:** Anna Spångeus, Catrin Willerton, Paul Enthoven, Ann-Charlotte Grahn Kronhed

**Affiliations:** 1Department of Acute Internal Medicine and Geriatrics, Linköping University Hospital, 581 83 Linköping, Sweden; 2Division of Diagnostics and Specialist Medicine, Department of Health, Medicine and Caring Sciences, Linköping University, 581 83 Linköping, Sweden; 3Rehab Väst, Local Health Care Services in the West of Region Östergötland, 592 32 Vadstena, Sweden; 4Division of Prevention, Rehabilitation and Community Medicine, Department of Health, Medicine and Caring Sciences, Linköping University, 581 83 Linköping, Sweden; 5Pain and Rehabilitation Centre, Department of Health, Medicine and Caring Sciences, Linköping University, 581 83 Linköping, Sweden

**Keywords:** chronic pain, rehabilitation, interdisciplinary pain treatment, primary health care, osteoporosis, patient education, quality of life, vertebral fracture

## Abstract

Fragility fractures, in particular vertebral fractures, are associated with high morbidity, including chronic pain and reduced health-related quality of life. We aimed to investigate the short- and long-term effects of patient education, including interdisciplinary themes, with or without physical training or mindfulness/medical yoga for patients with established spinal osteoporosis in primary care. Osteoporotic persons aged sixty years or older with one or more vertebral fractures were randomized to theory only, theory and physical exercise, or theory and mindfulness/medical yoga and were scheduled to once a week for ten weeks. Participants were followed up by clinical tests and questionnaires. Twenty-one participants completed the interventions and the one-year follow-up. Adherence to interventions was 90%. Pooled data from all participants showed significant improvements after intervention on pain during the last week and worst pain, and reduced painkiller use (any painkillers at baseline 70% [opioids 25%] vs. post-intervention 52% [opioids 14%]). Significant improvements were seen regarding RAND-36 social function, Qualeffo-41 social function, balance, tandem walking backwards, and theoretical knowledge. These changes were maintained at the 1-year follow-up. Patient group education combined with supervised training seems to have positive effects on pain, and physical function in persons with established spinal osteoporosis. The improved quality of life was maintained at the 1-year follow-up.

## 1. Background

Osteoporosis is defined as a systemic skeletal disease characterized by low bone mass and micro-architectural deterioration of bone tissue, leading to enhanced bone fragility and a consequent increase in fracture risk [1]. Fragility fractures, in specific vertebral and hip fractures, are associated with high morbidity, mortality and socioeconomic costs [2,3,4,5,6], and are also associated with an increased risk of new fractures, often in the near future [7,8]. Patients with a vertebral fracture (VF) often have lifelong and opioid-requiring pain as well as functional disabilities and reduced health-related quality of life (HRQoL) [3,6]. Thus, it is important to assess these patients for osteoporosis and to initiate adequate pharmacological treatment as well as to work on preventable risk factors, such as the risk of a fall (balance, functional disabilities, and home environment), nutrition, and adherence to treatment. Though well known, these adjustable risk factors are not systematically worked upon in many health organizations handling patients with osteoporosis. In most organizations, the primary health care provider is responsible for their osteoporosis patients [9] and organizing and securing this interdisciplinary teamwork.

Osteoporosis schools, i.e., patient education with an interdisciplinary focus, are part of some health organizations. The schools’ content, both regarding theory parts and possible physical training, vary as well as the timeframe and different patient categories included with regard to fracture history [10,11]. Theory content in the group’s education often focuses on knowledge of osteoporosis, medication, and nutrition [10]. Furthermore, many osteoporosis schools include physical activity in various arrangements concerning the frequency and specific activity. Though the results are inconsistent, probably because of heterogenous study designs and groups, some studies have shown a positive effect of patient education and training on patients’ HRQoL, and patient empowerment [10]. There is an obvious need for high-quality randomized controlled trials to evaluate the effectiveness of patient education combined with physical training in people with osteoporosis [12].

In the present study including patients with established spinal osteoporosis, we aimed to investigate the short- and long-term effects of interdisciplinary patient education with or without physical training or mindfulness and medical yoga on (1) chronic pain; (2) HRQoL; (3) physical strength and balance performance; (4) fall risk and physical activity; (5) theoretical knowledge; and (6) patient enablement.

## 2. Methods

### 2.1. Study Design

This pilot randomized study was called the School of Osteoporosis in Linköping (the SOL study) and included a 10-week intervention period with once-weekly theory education with or without additional physical training. Furthermore, a preceding observation period of 10 weeks, as well as a 1-year post-intervention follow-up, were included in the study design (Figure 1).

The study protocol has been registered at ClinicalTrials.gov, (ClinicalTrials.gov Identifier: NCT05227976) 12 January 2022. https://clinicaltrials.gov/ct2/show/NCT05227976 (assessed on 9 March 2023).

### 2.2. Participants

To be included in the SOL study, participants had to meet four criteria: (1) diagnosed with established spinal osteoporosis (at least one vertebral fracture and osteoporosis); (2) >3 months had passed since the most recent VF; (3) age ≥60 years; and (4) the physical ability to walk without an indoor walker. Patients with an inability to understand the Swedish language or difficulty following the research protocol, or dementia were excluded. The patients fulfilling the criteria are mostly followed up by the primary health care in Sweden.

### 2.3. Study Procedures

Participants were recruited by means of advertisements through the regional patient organization, local newspapers, primary health care centers and the osteoporosis unit (i.e., the unit performing dual-energy X-ray absorptiometry [DXA] scans for all patients in the region). A research nurse made the screening process by phone calls.

Clinical testing and a questionnaire-based evaluation were performed at baseline (May 2018), post-observation (Aug./Sept. 2018), and post-intervention (Nov./Dec. 2018) (Figure 1). The 1-year post-intervention follow-up (Nov./Dec. 2019) was based on a questionnaire, solely (Figure 1). The questionnaires were sent by postal mail and were answered at home. At post-intervention, the participants and therapists were asked about their experiences with the study procedures and interventions. The randomization was done blind by research staff after the baseline tests, where the subjects’ hidden names were drawn consecutively to one of the three groups, i.e., (1) theory only (T group); (2) theory and physical exercise (TPh group); and (3) theory and mindfulness/medical yoga (TMMY group).

### 2.4. Observation

During the four-month non-interventional observation period, participants were asked to live as usual. The data were analyzed as pooled data (*n* = 21), as no intervention had been done prior to the observation.

### 2.5. Interventions

All three intervention arms included the same theoretical lectures organized as a 1-h weekly theory session for 10 weeks. In addition, the TPh and TMMY groups had a 1-h training session scheduled in adjunction to the theory sessions, respectively. A coffee break was included as a social event for each meeting. The theory themes were (1) osteoporosis and physical activity (led by a physiotherapist); (2) diagnosis of osteoporosis and pharmacological treatment, lasting two sessions (led by a physician who specialized in endocrinology/osteoporosis); (3) mindfulness and medical yoga (led by a physiotherapist/MMY teacher); (4) orthopedic technician aspects of activating spinal orthosis and stable shoes (led by an orthopedic technician and a representative from an orthosis company); (5) nutritional aspects (led by a dietitian); (6) balance performance and balance training (led by a physiotherapist); (7) information from the regional patient association for osteoporosis (led by two representatives of the local patient organization); (8) ergonomic aspects concerning daily living activities and adequate technical support (led by an occupational therapist); and (9) physiology of pain (led by a member of a team working with interdisciplinary pain rehabilitation at Linköping University Hospital).

The theory sessions were organized by a moderator and were conducted in a conference room at Linköping University for the T group, whereas the TPh and TMMY groups had theory lessons together in a conference room at the training center. An experienced physiotherapist, with knowledge of appropriate training for persons with osteoporotic vertebral fractures, supervised the TPh group for 45 min once a week prior to the theory sessions. The exercise program started with a warm-up phase for six minutes and was followed by circuit training (performed by standing or walking) at nine training stations focusing on muscle strength and balance exercises for forty-five seconds times three sets. The sessions ended with a 5 min cool-down and stretching. All participants in the TPh group also received a home training program. The TMMY sessions were equally scheduled prior to the theory sessions and started with thirty-minute modified medical yoga exercises and yoga meditations (sitting on comfortable chairs) and ended with one leg standing. The yoga poses for the back were modified to suit each participant’s individual needs and there were neither extreme positions, strenuous spinal flexion exercises, exercises with rotation nor exercises bending the trunk to an end-range position [13]. Furthermore, the TMMY group practiced the mindfulness concept for another 30 min with mainly breathing and awareness exercises. Mindfulness theory as well as weekly training follow-ups were part of the concept, and the participants also received a CD with mindfulness awareness exercises on their first session for daily home practice. An experienced physiotherapist, who was also a mindfulness/medical yoga instructor, supervised the TMMY sessions [14].

### 2.6. Outcomes

The patient-reported questionnaires consisted of two parts: (1) a self-constructed questionnaire including clinical background data, present medication, including pain killers, history of falls, self-estimated disease knowledge, and performed physical activity; and (2) validated instruments about pain, HRQoL, fall risk evaluation, physical activity, and patient enablement.

#### 2.6.1. Pain

Present pain, pain from last week, and worst pain were estimated using a numeric rating scale (NRS), ranging from 0 = no pain to 10 = worst possible pain at the clinical testing occasions. Furthermore, patients were asked about their usage of painkillers (including the type and regularity). Effects on pain were also assessed by the HRQoL instruments (below).

#### 2.6.2. Health-Related Quality of Life

The measurement of HRQoL included generic instruments (the European quality of life, EQ-5D-3L, and RAND-36), as well as a disease-specific (established spinal osteoporosis) instrument, the “quality of life questionnaire in the European Foundation for Osteoporosis-41” (Qualeffo-41) [15,16,17,18]. The EQ-5D-3L comprises five dimensions i.e., mobility, self-care, usual activities, pain/discomfort, and anxiety/depression. These dimensions are combined with an EQ-5D-3L index normalized to a reference ranging from −0.594 to 1, where 1 indicates optimal health [15]. The RAND-36 comprises 36 items organized into eight different health domains i.e., physical function (PF), role physical (RP), bodily pain (BP), general health (GH), vitality (VT), social function (SF), role emotional (RE), and mental health (MH) [16,17]. The scores were transformed into a 0–100 scale (0 = worst possible HRQoL and 100 = the best possible). The Qualeffo-41 includes 41 questions summarized into seven domains: (1) pain (backache); (2) activities of daily living (ADL); (3) jobs (around the house); (4) mobility; (5) social function; (6) general health perception; and (7) mental function [18,19]. In contrast to RAND-36, 0 indicates the best possible and 100 the worst possible HRQoL.

#### 2.6.3. Physical Strength, Balance Performance, and Anthropometry

Static and dynamic balance tests were performed, including the one-leg standing (right and left) tests that were done with the opposite foot positioned on the calf of the tested leg and the arms along the sides with the eyes open and then closed, respectively, and were limited to a maximum of 30 s. The dynamic balance tests were done by tandem walking heel-to-toe forwards, and toe-to-heel backwards on a line. The number of steps were counted and maximized to 15 correct steps. The balance tests were performed three times and the best trial was used as the final score [20]. In the chair-stand test, the participants were asked to rise as many times as possible from a standard chair with knees bent at a 90° angle for 30 s without the assistance of the arms [21]. Grip strength (kg) of the dominant and the non-dominant hand was measured by the standard Jamar dynamometer. Each test was performed three times and the best trial was used [22]. The distance (cm) between the seventh cervical vertebra (C7) and the wall was measured by a folding ruler to estimate the back-straightening ability [23]. Body height was measured with a stadiometer and body weight with a digital scale. All clinical tests were performed by a physiotherapist with extensive clinical and research experience and great familiarity with the tests.

#### 2.6.4. Fall Risk and Physical Activity

The Swedish Falls Efficacy Scale International (FES-I) comprises 16 items about the risk of falling (including a four-level scale, where 1 = not at all concerned and 4 = very concerned). The individual scores were summarized to a total score [24]. The questionnaire also comprised questions about physical exercise, physical activity, and sedentary behavior [25].

#### 2.6.5. Theoretical Knowledge Assessment

To evaluate the effect of the patients’ education, their knowledge of osteoporosis was assessed by ten open-ended questions at baseline and post-intervention. The questions tested a basic knowledge of osteoporosis, medication, exercise and fall prevention, and were produced and evaluated by experienced university teachers. The total score was between 0 (worst knowledge) and 22 (best knowledge).

#### 2.6.6. Patient Enablement and Overall Experiences of the SOL

The Patient Enablement Instrument (PEI) was used to measure the extent to which a patient can understand and cope with his or her illness after an intervention [26]. The PEI consisted of six questions starting with “As a result of the osteoporosis school, do you feel you are …” followed by the alternative answers, where score 2 was for much better/much more; score 1 was for better/more; and score 0 was for same, less, or not applicable. The total score (range 0–12) was calculated for those participants who answered at least five questions. A higher score indicated higher enablement [26]. The Swedish version has been shown to have acceptable validity and reliability for patients with chronic pain [27,28].

The participants’ overall experiences of the theoretical lectures and the physical training were scored on a six-level scale (where 5 was considered very satisfied and 0 was not at all satisfied) post-intervention.

## 3. Statistical Analyses

Descriptive statistics at the different time points were reported with the median (Md) and interquartile range (IQR 25–75%), mean (M) and standard deviation (SD), and numbers and percentages. The Wilcoxon signed-rank test was chosen for comparing the change in each group over time. The relationships between the HRQoL measures RAND-36 GH, Qualeffo-41 GH and the total score, and the EQ-5D index, were investigated with Spearman’s rank correlation, using the following coefficients: 0–0.25 none to little; 0.25–0.50 fair; 0.50–0.75 moderate to good; >0.75 very good to excellent [29]. All statistical tests were performed at the 5% significance level. For the statistical analyses, SPSS 25.0 software (IBM Statistics, New York, NY, USA) was used.

## 4. Results

The inclusion period was set to four weeks. Sixty-two persons were interested in participating in the study and were contacted, but 50% (*n* = 31) did not meet the inclusion criteria. The most common reason was that they had no vertebral fracture (*n* = 23). An appropriate number of ten persons per group was set before the inclusion. However, as a total of thirty-one participants (two men and twenty-nine women) met the inclusion criteria during the enrollment period, all were included and randomized to either the T group (10 participants), TPh group (11 participants), or TMMY group (10 participants). Five people dropped out immediately after randomization, where three drop-outs were in the T group, one in the TPh group and one in the TMMY group. The reasons given for non-participation were the disappointment in being randomized to the T group, economic reasons, transportation to the education sessions, scheduled surgery, and serious illness of a family member. Another person in the TMMY group dropped out from the initiated intervention due to relocating to a new residence. Thus, the total number of study participants for the intervention period was 25. Several participants lived in other places than Linköping, at most 70 km from the SOL. There was a high attendance rate to the intervention program, with an average of nine out of ten (range 6–10) participants attending the SOL sessions. Twenty-one subjects (84%, 21/25) participated in the 1-year post-intervention follow-up. The four participants who dropped out of the one-year post-intervention follow-up were two participants from the T group; one from the TPh group; and one from the TMMY group. In the present study, only those 21 participants with complete data from baseline to the 1-year post-intervention follow-up were analyzed.

### 4.1. Background Characteristics

The baseline median (range) age for all participants (twenty women and one man) was seventy-two years (60–82). The median age was 72 years (67–82) in the T group (*n* = 5), 72 years (60–81) in the TPh group (*n* = 9), and 72 years (63–81) in the TMMY group (*n* = 7), without significant differences between the groups. Fifty-six percent of the participants had a history of fractures in addition to vertebral fractures. The most common fracture was the distal forearm reported by four participants. The majority of participants (70%) used regular painkillers (any type). Opioids were used regularly by 25% of participants. Most participants were clinically considered to have primary osteoporosis. Some participants had risk factors that might also affect the skeleton (secondary osteoporosis). This included six patients with thyroid disease and levothyroxine medication, one patient with chronic obstructive lung disease, and one person with previous breast cancer. In the final cohort (*n* = 21), there were no participants using continuous glucocorticoids or with a known inherited bone metabolic disease. At the baseline, 62% had an ongoing antiresorptive treatment (alendronic acid [*n* = 2], zoledronic acid [*n* = 8], and denosumab [*n* = 5]). No patients had an ongoing bone anabolic treatment. Calcium and/or vitamin D supplements were taken by 95% of participants. The present study focused on elderly community dwellers in primary care. No participants said that they had a community-based home help service. All participants were independent of indoor walking aids when included, as one of our exclusion criteria was a dependency on such aids. One person in each group was dependent on outdoor walking aids and some also used walking poles to support their balance during walks.

### 4.2. Observation Period

During the non-interventional observation period, no changes were seen regarding pain from last week, worst pain, HRQoL (RAND-36, Qualeffo-41, and EQ-5D), fall risk (FES-I), and physical activity (Appendix A). Significant improvements were seen regarding distance C7-wall (Md 6.5 cm vs. 6 cm, *p* = 0.025) and the chair-stand test (Md 9 vs. 11, *p* = 0.002), but not in the other clinical tests.

### 4.3. Intervention and Long-Term Effect

#### 4.3.1. Pain

The NRS score for “worst pain” improved from 7.8 (Md) at baseline to 6.3 post-intervention (*p* = 0.013), and “pain last week” improved from 5 (Md) to 4.5 (*p* = 0.042) when analyzing the pooled data of all participants (Table 1). When analyzing the separate intervention groups, the TPh group showed close to significant improvement regarding NRS scores for both variables (*p* = 0.058 and *p* = 0.106). This pilot study was, however, not powered for subgroup analysis, which is why its significance should be interpreted cautiously, and its trends should be verified in larger cohorts.

The percentage of all participants using opioids decreased from 25% (baseline) to 14% (post-intervention) and increased to 19% again at the 1-year post-intervention follow-up (Figure 2). About two-thirds (70%) of the participants used painkillers regularly at baseline. After the intervention, this figure improved to 48% and remained at that level at the 1-year follow-up.

#### 4.3.2. Health-Related Quality of Life

The pooled data of all the participants showed a significantly improved scoring of social function (*p* = 0.048), and close to significant scoring of physical function (*p* = 0.060) and mental health (*p* = 0.071) post-intervention compared to the baseline for RAND-36 (Table 2). No change was seen regarding RAND-36 dimensions at the 1-year post-intervention follow-up compared to post-intervention. When analyzing the separate intervention groups, the TPh group showed a significantly improved score regarding physical function (*p* = 0.041) and close to significant for bodily pain (*p* = 0.063). The TMMY group showed a significantly improved mental health score (*p* = 0.040) and close to significant improvement in social function (*p* = 0.066). No significant change was seen in the T group. These results should be interpreted cautiously, as the number of participants was very small.

Regarding the Qualeffo-41, the social function domain was improved post-intervention (*p* = 0.024) in the pooled data for all participants (Table 3). At the 1-year post-intervention follow-up, no change was seen in any Qualeffo-41 domain compared to post-intervention. For the separate intervention group analysis, the pain improved post-intervention in the TPh group (*p* = 0.046), the social function improved in the TMMY group (*p* = 0.043) and the mental health score almost significantly improved in the T group (*p* = 0.068).

No significant change was seen when analyzing the pooled data from EQ-5D neither at the baseline versus post-intervention nor post-intervention versus the 1-year post-intervention follow-up. Only the TPh group showed a trend of improved EQ-5D post-intervention (*p* = 0.068).

#### 4.3.3. Physical Strength, Balance Performance, and Anthropometry

The pooled data from all participants showed improved values post-intervention regarding the chair-stand test (*p* = 0.005), one-leg stand using the left leg with the eyes closed (*p* = 0.030), and tandem walking backwards (*p* = 0.027) (Table 4). The one-leg stand on the right leg with the eyes open showed worse results post-intervention when pooling data (*p* = 0.019). However, there was no change when analyzing the separate intervention groups.

The analysis of the intervention groups showed post-intervention balance improvements mostly in the TPh group, such as tandem walking forwards (close to significant *p* = 0.068), one-leg stance with eyes closed (right leg *p* = 0.040 and left leg *p* = 0.026) and close to significant improvements on the chair-stand test (*p* = 0.075). The TMMY group showed improved chair-stand test results post-intervention (*p* = 0.018).

No change was seen regarding handgrip strength, C7-wall distance, body weight, or height.

#### 4.3.4. Fall Risk and Physical Activity

No change was seen in FES-I scoring neither post-intervention nor at the 1-year post-intervention follow-up (Table 5).

There were five participants both at the baseline and post-intervention who reported physical activity of less than 150 min/week. At the 1-year post-intervention follow-up, six people reported insufficient physical activity (one person had recently suffered from a calcaneus fracture). There was no significant change in the reported total physical activity in the intervention groups between the baseline and post-intervention or between the post-intervention and 1-year post-intervention follow-up. However, a trend of more physical activity was seen in the T group (*p* = 0.066). The sedentary inactive time (sitting/resting) did not change between the baseline and follow-ups.

#### 4.3.5. Theoretical Knowledge

The participants scored significantly better in the knowledge testing about osteoporosis post-intervention compared to the baseline. The median knowledge score increased from 15 at the baseline to 18 post-intervention (68% vs. 81% correct answers, *p* = 0.001), when analyzing the pooled data of all participants. The median score increased from 15 to 19 (68% vs. 86%, *p* = 0.043) in the T group, from 11 to 18 (50% vs. 81%, *p* = 0.011) in the TPh group, and from 17 to 20 (77% vs. 91%, *p* = 0.018) in the TMMY group.

#### 4.3.6. Patient Enablement and Overall Experiences of the SOL

The mean (SD) PEI total score post-intervention was 4.6 (2.8) (Md = 5, range 0–9). The mean (SD) values for the groups were 3.0 (2.9) for the T group, 3.6 (2.9) for the TPh group, and 5.8 (2.4) for the TMMY group. Improved patient enablement, i.e., “better” or “much better”, was reported by 40–92% of participants in each question, as shown in Figure 3. The question with the highest percentage of participants reporting improved enablement was “(Q2)… able to understand your illness/disabilities” (92% of participants), and the question with the lowest improvement was “(Q5)… confident about your health” (40% of participants).

The mean (SD) PEI total score at the 1-year follow-up was 4.1 (3.4) (Md = 4, range 0–11). The mean (SD) values for the groups were 3.8 (3.8) for the T group, 2.7 (2.6) for the TPh group, and 6.3 (3.1) for the TMMY group. The question with the highest percentage of participants reporting improved enablement was “(Q2)… able to understand your illness/disabilities” (76% of participants), and the question with the lowest improvement was “(Q5)… confident about your health” (33% of participants). The PEI total score did not change significantly between the post-intervention and 1-year post-intervention follow-up (*p* = 0.390).

The theoretical lectures were generally appreciated by the participants, scoring a mean of 4.4 (range 4.0–4.6) and a median of 5 (range 4.5–5). Similarly, the participants in the physical intervention groups scored highly regarding their satisfaction with the mindfulness/medical yoga activities (mean 4.7/Md 5), and also the physical training activities (mean 4.9/Md 5), respectively.

#### 4.3.7. Adverse Events

One participant in the TMMY group reported a rib fracture just before the start of the intervention. There was one participant in the TPh group who had a finger fracture during the group training activities.

At the 1-year post-intervention follow-up, a vertebral fracture was reported in the T group, a calcaneus fracture (after falling from a ladder) was reported in the TPh group, and a rib fracture and a sacral fracture were reported in the TMMY group. At the baseline, one participant in the TPh group reported a fall in the past month, while one participant in the T group reported a fall at the end of the intervention period. No participants reported a low energy fall at the 1-year post-intervention follow-up.

## 5. Discussion

The present study shows that patient education, including interdisciplinary themes, combined with supervised training is a well-accepted and safe intervention in a primary health care cohort of patients with established spinal osteoporosis. The intervention had positive effects on important health outcomes, including chronic pain, physical function, and HRQoL. Adherence to the once-weekly intervention program for ten weeks was high, with a mean attendance rate of 90%, even though several participants lived far away from where the SOL interventions took place.

Most VFs are painful and cause persistent symptoms [3,6,30]. In our study, one out of four participants reported the use of opioids at the baseline, and more than half of the group scored more than five on the NRS as the worst pain, thus highlighting the impact of the pain. After the intervention, regular usage of opioids as well as other painkillers decreased, and participants reported less pain (last week and worst). Physical training might decrease pain [31], which agrees with our finding that the TPh group showed improvements in the HRQoL pain dimensions after the intervention period. This strengthens the need for physical training as part of the education program for patients with established spinal osteoporosis [32]. Coping with pain could also be influenced by increased knowledge of the disease and by shared experiences with other participants in a similar situation [33].

Concerning the importance of observing and teaching the individuals within the groups, an upper limit of 8 to 10 participants is desirable [14,34]. At the baseline, there were 10–11 randomized participants in each intervention group. However, the number decreased below ten because of the dropouts.

A multi-component exercise program with progressive strength resistance and balance training, which was performed by the TPh group, is recommended for people with osteoporosis [34,35,36]. The SOL was scheduled to be once a week for practical reasons, but 10 supervised 1-h group training sessions seemed not long enough to observe some changes, such as improved body posture. Indeed, resistance and balance training at least twice a week is recommended to achieve effects on falls and bone health [35].

Most of the physical exercise training and the MMY training was safe. However, one finger fracture occurred in the TPh group, when exercising with a medicine ball. This specific exercise could be adjusted in the future to diminish the risk of injuries in this patient group.

Previous studies on patient education in osteoporosis have shown inconsistent results on quality of life, maybe due to various interventions and study designs [10,11,12]. Bergland et al. showed that a combined 3-month course of circle exercise (biweekly sessions) and a 3-h theory session had a positive effect on HRQoL in patients with vertebral fractures, which was still seen after 12 months [37]. On the contrary, Kessenich et al. showed a minimal effect on HRQoL after an 8-week educational support group, along with weekly telephone calls, in women with established spinal osteoporosis. However, that study, which did not include any supervised group training activities (besides Tai Chi exercises introduced by a speaker), proposed a combination of rehabilitative approaches to improve the quality of life for the patients [38]. In the present study of patient education (osteoporosis school) for established spinal osteoporosis, once-a-week sessions for 10 weeks seemed to be most effective when the theoretical lectures were followed by physical training. The importance of providing patient education to patients with osteoporosis is supported by recent guidelines from the Swedish National Board of Health and Welfare giving patient education a rather high priority [39].

The one-leg stance with eyes closed was improved for all participants as a group but, when the subgroups were analyzed, only the TPh group had significant improvements in this test. However, there was no improvement in the one-leg stance with the eyes open. We experienced that the limit for the one-leg stance with the eyes open could be extended to 60 s, as many of the participants managed 30 s, which could make the outcome measure less sensitive. The chair-stand was improved in the TMMY group, maybe as a result of an actual improvement or a more confident patient after the supervised mindfulness/medical yoga training [14]. Improved mental health and social function were also found in the TMMY group.

To our knowledge, this is the first study using the PEI to measure patient enablement in patients with osteoporosis. The mean PEI total score with a range of 0-12 was 4.1 post-intervention, which is a similar improvement compared to another Swedish study investigating patient enablement in patients with chronic musculoskeletal pain after intervention [27].

The power of this pilot study makes subgroup analysis uncertain, but some trends could be seen, including the additive effect when including physical training in the patients’ education. However, the pure T group was very small, with a risk of rejecting true improvements. With this perspective, it is still important to include a pure T group when designing RCTs. In addition, theory education could more easily be implemented through a digital osteoporosis school. A sample size calculation with “pain last week” as the primary outcome and a 20% reduction after an intervention show a need for at least 17 participants in each arm.

In the study design, we also included a non-interventional observation period prior to the interventions. We initially intended to conduct the study with a control arm without patient education interventions, i.e., similar to how osteoporosis health care is arranged in many areas today. However, after some critical reviews from the ethical committee, we chose to let all participants have an active arm and added the preceding observation period. Analyses of these pooled data showed stable results regarding most variables strengthening the observations that the changes seen after intervention are effects of the SOL. In the observation period, the distance C7-wall was improved but did regress to the baseline values after the interventions. We believe that this was not a true change but a weakness of the test, i.e., too little precision from using a folding ruler to make the correct measurements. The general unaltered results during the observation period indicate that a non-interventional arm might be unnecessary for a future larger RCT.

The informants in the TPh group, as well as participants in the other intervention groups, asked for supervised physical group training after the SOL, which was planned to begin in January 2020. Unfortunately, this training was impeded due to the lockdown caused by the COVID-19 pandemic.

### Study Strengths and Limitations

The present SOL study is a pilot study with an RCT design, where patients were randomized to one of three different interventions, all with a common theoretical base. A limitation of our study is the small number of participants who completed both the intervention and the 1-year post-intervention follow-up (*n* = 21).

## 6. Conclusions

The present pilot study showed that interdisciplinary patient group education, with or without supervised exercise or mindfulness/modified medical yoga, might improve short-term physical function and pain, and short- and long-term quality of life and patient enablement in patients with established spinal osteoporosis in primary care. More randomized controlled studies are needed with a higher number of participants to confirm our results.

## Figures and Tables

**Figure 1 ijerph-20-04933-f001:**
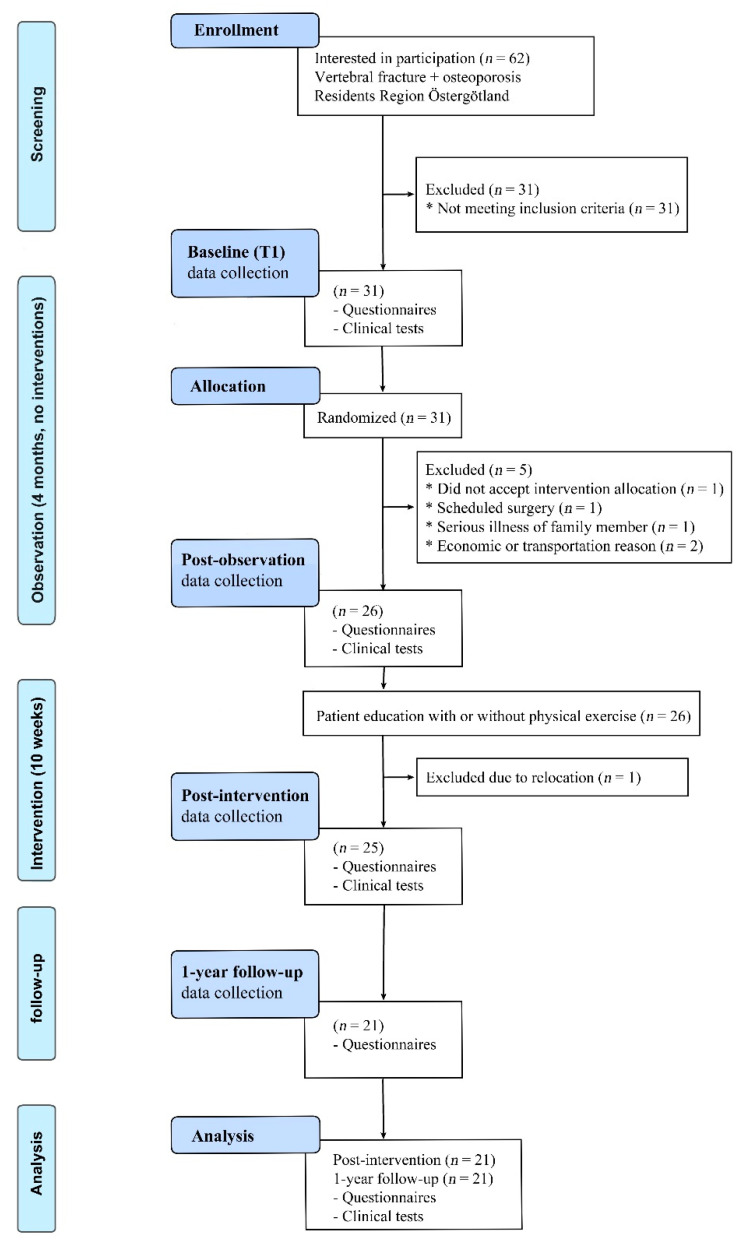
CONSORT flow chart. Patients were tested clinically and evaluated by questionnaires at four time points: (1) at baseline; (2) after a passive observation time of 4 months; (3) post-intervention of patient education including weekly theory sessions for 10 weeks with or without additional group training; and (4) at the 1-year follow-up after the intervention. In the current study, only participants with data from baseline, post-intervention and the 1-year post-intervention follow-up were included (*n* = 21). Regarding the T group: 10 participants were randomized at baseline, 7 attended intervention, and 5 attended the 1-year post-intervention follow-up. Similar figures for the TMMY group were: 10 randomized, 8 at intervention, and 7 at the 1-year post-intervention follow-up. The TPh group figures were: 11 randomized, 10 at intervention, and 9 at the 1-year post-intervention follow-up.

**Figure 2 ijerph-20-04933-f002:**
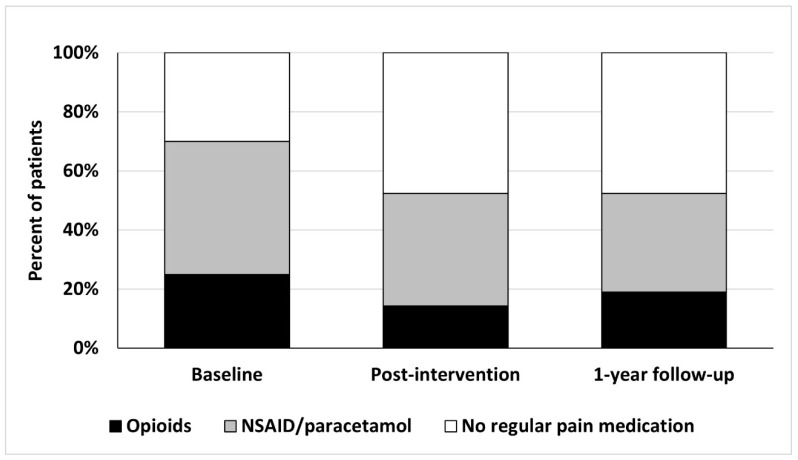
Usage of analgesics during the three different time points. Patients on both opioids and non-steroid anti-inflammatory drugs (NSAID) or paracetamol were included only in the opioid group.

**Figure 3 ijerph-20-04933-f003:**
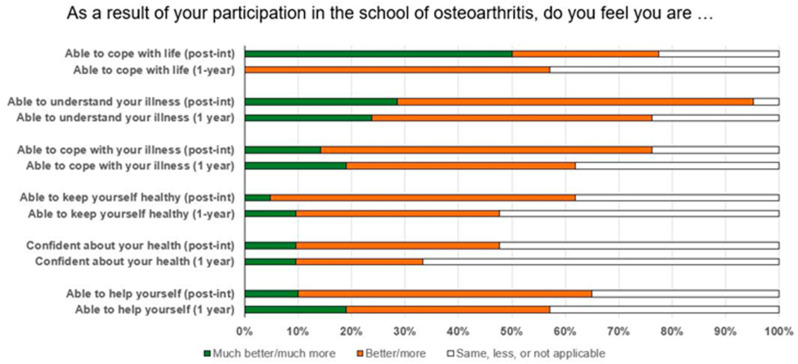
Patient enablement instrument (PEI) pooled data of all participants at the post-intervention and 1-year post-intervention follow-up.

**Table 1 ijerph-20-04933-t001:** Pain assessed by a numeric rating scale (NRS) at baseline and post-intervention.

	Baseline Md (X) 25–75%	Post-intervention Md (X) 25–75%	*p*-Value
**Current pain, NRS, All**	3.5 (3.2) 0.3–5.3	1.0 (1.9) 0.0–4.3	0.093
T group	2.5 (3.5) 1.0–6.5	4.5 (3.0) 0.5–4.8	1.000
TPh group	4.0 (3.7) 1.8–5.0	0.0 (1.6) 0.0–3.3	0.138
TMMY group	0.0 (2.4) 0.0–6.0	0.0 (1.6) 0.0–3.5	0.500
**Pain last week, NRS, all**	5.0 (5.2) 4.0–6.9	4.5 (3.7) 0.5–5.9	**0.042**
T group	8.0 (5.9) 2.3–8.5	4.5 (5.2) 4.3–6.5	0.498
TPh group	4.5 (4.9) 4.0–6.3	2.3 (2.7) 0.0–5.3	0.106
TMMY group	5.0 (5.1) 4.0–6.0	5.0 (3.8) 0.0–6.0	0.271
**Worst pain, NRS, all**	7.8 (7.3) 6.3–8.3	6.3 (6.2) 4.9–8.0	**0.013**
T group	8.0 (8.1) 6.8–9.5	7.5 (7.1) 5.8–8.3	0.221
TPh group	6.5 (6.4) 5.0–8.0	5.5 (5.0) 3.5–5.5	0.058
TMMY group	8.0 (7.8) 6.9–9.0	6.8 (6.8) 5.8–8.0	0.223

Abbreviations: Md, median; and (X), mean. Bold text: *p* < 0.05. All *n* = 21; T group *n* = 5, TPh group *n* = 9; and TMMY *n* = 7.

**Table 2 ijerph-20-04933-t002:** Health-related quality of life and the change between the baseline, post-intervention, and 1-year post-intervention follow-up (*RAND-36*).

		Baseline (1) Md (X) 25–75%	Post-intervention (2) Md (X) 25–75%	1-Year Follow-Up (3) Md (X) 25–75%	*p*-Value (1) vs. (2)	*p*-Value (2) vs. (3)
**Health-related quality of life**						
	*RAND-36*					
	**Physical function PF, all**	70 (60) 35–85	70 (64) 43–85	60 (59) 40–85	0.060	0.146
	T group	60 (53) 23–80	70 (56) 25–80	60 (51) 13–85	0.180	0.257
	TPh group	70 (62) 40–83	80 (70) 53–88	65 (66) 43–85	**0.041**	0.344
	TMMY group	70 (62) 40–85	70 (62) 40–80	60 (56) 35–85	0.713	0.344
	**Role physical RP, all**	25 (39) 0–75	38 (49) 0–100	38 (49) 0–100	0.309	1.000
	T group	25 (25) 0–50	25 (40) 0–88	0 (40) 0–100	0.317	1.000
	TPh group	50 (42) 0–75	100 (64) 13–100	50 (58) 25–100	0.143	0.414
	TMMY group	50 (50) 0–100	0 (25) 0–75	25 (38) 0–88	0.317	0.317
	**Bodily pain BP, all**	45 (49) 45–63	58 (55) 39–78	45 (52) 33–68	0.163	0.443
	T group	58 (42) 12–64	58 (39) 12–58	33 (38) 23–57	0.785	0.786
	TPh group	45 (48) 45–47	68 (64) 45–80	58 (60) 45–79	0.063	0.610
	TMMY group	55 (56) 35–78	45 (55) 33–78	55 (50) 25–68	0.786	0.916
	**General health GH, all**	55 (52) 28–73	65 (57) 35–73	55 (54) 35–70	0.203	0.154
	T group	50 (46) 15–76	65 (53) 28–73	55 (52) 30–73	0.345	0.891
	TPh group	55 (54) 25–78	60 (58) 35–80	55 (56) 33–73	0.608	0.573
	TMMY group	50 (55) 45–70	65 (59) 35–65	50 (53) 35–65	0.496	0.098
	**Vitality VT, all**	55 (55) 33–73	60 (58) 38–78	60 (60) 43–75	0.483	0.776
	T group	40 (49) 35–68	55 (52) 28–75	60 (53) 35–68	1.000	1.000
	TPh group	45 (50) 28–70	65 (55) 28–80	50 (62) 43–85	0.232	0.248
	TMMY group	65 (65) 55–80	60 (65) 50–75	60 (61) 55–70	0.932	0.416
	**Social function SF, all**	75 (75) 50–100	100 (83) 63–100	88 (80) 63–100	**0.048**	0.476
	T group	88 (78) 50–100	100 (78) 44–100	75 (73) 44–100	1.000	0.414
	TPh group	63 (74) 57–100	100 (85) 63–100	88 (85) 69–100	0.136	0.915
	TMMY group	75 (73) 50–100	88 (84) 63–100	88 (80) 50–100	0.066	0.285
	**Role emotional RE, all**	100 (65) 33–100	100 (76) 50–100	100 (67) 17–100	0.262	0.395
	T group	33 (40) 0–84	100 (60) 0–100	0 (33) 0–84	0.414	0.180
	TPh group	100 (67) 33–100	100 (78) 50–100	100 (81) 67–100	0.461	0.785
	TMMY group	100 (100) 33–100	100 (100) 50–100	100 (78) 33–100	1.000.	0.317
	**Mental health MH, all**	76 (73) 66–88	80 (77) 66–92	80 (75) 60–88	0.071	0.269
	T group	68 (66) 52–80	80 (74) 50–94	60 (66) 56–80	0.343	0.416
	TPh group	80 (72) 52–86	72 (72) 52–88	80 (73) 58–88	0.765	0.596
	TMMY group	76 (79) 68–88	88 (86) 76–92	88 (83) 76–88	**0.040**	0.389

Abbreviations: Md, median; and (X), mean. Bold text: *p* < 0.05. All *n* = 21; T group *n* = 5, TPh group *n* = 9; and TMMY *n* = 7.

**Table 3 ijerph-20-04933-t003:** Health-related quality of life and the change between the baseline, post-intervention, and 1-year post-intervention follow-up (*Qualeffo-41*).

		Baseline (1) Md (X) 25–75%	Post-Intervention (2) Md (X) 25–75%	1-Year Follow-Up (3) Md (X) 25–75%	*p*-Value (1) vs. (2)	*p*-Value (2) vs. (3)
**Health-related quality of life**						
	*Qualeffo-41*					
	**Pain, all**	55 (51) 35–65	45 (46) 30–67	40 (46) 28–68	0.143	0.615
	T group	60 (55) 33–75	65 (56) 38–70	50 (57) 38–80	0.786	0.854
	TPh group	60 (56) 45–70	40 (45) 35–53	35 (41) 23–65	**0.046**	0.865
	TMMY group	40 (42) 25–55	38 (40) 30–69	40 (44) 35–65	0.865	0.270
	**Activities of daily life, all**	13 (19) 6–25	19 (18) 6–19	13 (17) 6–22	0.709	0.326
	T group	19 (25) 9–44	13 (20) 6–38	19 (25) 9–44	0.581	0.461
	TPh group	13 (16) 6–19	19 (15) 6–19	13 (12) 6–16	0.730	0.096
	TMMY group	13 (18) 6–31	19 (19) 13–31	19 (17) 6–25	0.713	0.257
	**Jobs around the house, all**	20 (27) 10–45	25 (29) 5–50	15 (25) 5–48	0.417	0.075
	T group	20 (30) 10–55	30 (37) 5–73	35 (31) 3–58	0.257	0.705
	TPh group	10 (18) 5–33	5 (17) 5–35	10 (13) 3–25	0.792	0.107
	TMMY group	45 (37) 10–55	50 (39) 15–55	45 (35) 10–55	0.683	0.339
	**Mobility, all**	22 (25) 9–46	19 (25) 11–38	22 (25) 11–33	0.659	0.587
	T group	22 (31) 19–48	28 (35) 19–55	28 (35) 13–61	0.180	0.854
	TPh group	25 (25) 13–38	19 (22) 11–38	22 (23) 13–33	0.234	0.596
	TMMY group	9 (21) 6–46	13 (21) 6–38	16 (20) 6–31	0.684	0.496
	**Social function, all**	40 (41) 22–61	33 (34) 17–49	37 (37) 15–58	**0.024**	0.509
	T group	53 (55) 43–67	43 (49) 33–67	43 (51) 36–69	0.500	0.893
	TPh group	22 (33) 16–61	20 (30) 15–49	19 (28) 12–44	0.484	0.441
	TMMY group	40 (42) 26–66	31 (28) 8–43	29 (38) 17–66	**0.043**	0.108
	**General health perception, all**	58 (56) 38–71	58 (52) 42–67	58 (53) 38–71	0.208	0.916
	T group	58 (67) 50–88	67 (67) 54–79	58 (58) 33–83	1.000	0.276
	TPh group	50 (52) 29–75	50 (46) 17–67	58 (52) 33–67	0.205	0.480
	TMMY group	58 (52) 33–67	58 (50) 42–58	58 (51) 42–67	0.465	0.893
	**Mental function, all**	36 (39) 31–50	42 (40) 22–57	36 (46) 26–54	0.695	0.295
	T group	44 (53) 36–75	33 (41) 25–61	53 (47) 25–67	0.068	1.000
	TPh group	33 (34) 22–50	22 (34) 21–49	36 (47) 24–50	0.766	0.342
	TMMY group	36 (35) 28–47	53 (46) 19–67	33 (45) 31–56	0.352	0.866
	**Total score, all**	32 (36) 25–48	30 (34) 24–45	36 (34) 20–44	0.259	0.578
	T group	35 (45) 30–64	36 (42) 26–62	41 (42) 22–63	0.500	0.893
	TPh group	28 (32) 25–42	30 (29) 20–40	25 (30) 20–42	0.192	0.678
	TMMY group	36 (34) 21–45	37 (34) 24–46	38 (34) 17–45	0.612	0.612
	**EQ-5D, all**					
	EQ-5D index	0.73 (0.63) 0.62–0.80	0.80 (0.72) 0.63–0.80	0.75 (0.66) 0.66–0.80	0.138	0.192
	T group	0.73 (0.67) 0.57–0.75	0.80 (0.62) 0.35–0.80	0.73 (0.52) 0.14–0.80	0.715	0.593
	TPh group	0.80 (0.54) 0.11–0.80	0.80 (0.76) 0.71–0.80	0.80 (0.74) 0.66–0.80	0.068	0.180
	TMMY group	0.69 (0.74) 0.62–0.85	0.73 (0.75) 0.62–0.85	0.75 (0.65) 0.54–0.80	0.655	0.465

Abbreviations: Md, median; (X), mean; Qualeffo, quality of life questionnaire in the European Foundation for Osteoporosis; EQ-5D, European quality of life 5 dimensions; and NRS, numeric rating scale. Bold text: *p* < 0.05. All *n* = 21; T group *n* = 5; TPh group *n* = 9; and TMMY *n* = 7.

**Table 4 ijerph-20-04933-t004:** Clinical testing outcomes and the change between the baseline and post-intervention.

		Baseline (1) Md (X) 25–75%	Post-intervention (2) Md (X) 25–75%	*p*-Value (1) vs. (2)
**Clinical tests**				
	**Distance C7-wall (cm), all**	6.5 (7.9) 5–11	6.5 (7.5) 5–10	0.225
	T group	7 (9.8) 6–15	7.5 (10) 5.8–15.5	0.854
	TPh group	8 (8.1) 5–11	7 (7.7) 5.3–10.3	0.429
	TMMY group	6 (6.2) 4–6.5	5.5 (5.6) 4–6.5	0.167
	**Hand force right, all**	20 (20) 17–24	21 (21) 17–25	0.667
	T group	16 (19) 14–26	20 (19) 12–25	0.785
	TPh group	20 (21) 18–23	21 (21) 17–24	0.596
	TMMY group	23 (21) 18–27	23 (21) 18–26	0.257
	**Hand force left, all**	18 (19) 14–23	18 (20) 15–23	0.913
	T group	13 (16) 11–22	20 (18) 11–24	0.416
	TPh group	18 (21) 17–24	18 (20) 16–22	0.524
	TMMY group	19 (20) 16–23	18 (19) 15–24	0.726
	**Chair-stand test, all**	9 (9) 7–11	13 (12) 11–15	**0.005**
	T group	9 (9) 6–11	12 (10) 5–15	0.492
	TPh group	8 (10) 7–13	13 (13) 11–17	0.075
	TMMY group	9 (9) 7–10	13 (13) 12–15	**0.018**
	**Right leg, eyes open (s), all**	30 (23) 15–30	14 (18) 7–30	**0.019**
	T group	30 (26) 17–30	11 (15) 7–26	0.109
	TPh group	18 (18) 10–30	13 (14) 4–25	0.208
	TMMY group	30 (28) 30–30	30 (24) 11–30	0.180
	**Left leg, eyes open (s), all**	27 (21) 12–30	18 (20) 10–30	0.396
	T group	17 (19) 10–29	24 (21) 10–30	0.285
	TPh group	18 (18) 8–30	15 (16) 7–24	0.340
	TMMY group	30 (27) 30–30	30 (25) 18–30	0.180
	**Right leg, eyes closed (s), all**	3 (4) 2–5	3 (4) 2–6	0.289
	T group	2 (3) 2–7	3 (4) 2–6	0.785
	TPh group	2 (2) 1–4	3 (3) 2–4	**0.040**
	TMMY group	4 (5) 3–6	4 (5) 3–7	0.752
	**Left leg, eyes closed (s), all**	2 (3) 1–4	3 (5) 2–6	**0.030**
	T group	2 (3) 2–4	2 (4) 1–10	0.593
	TPh group	3 (2) 1–3	4 (6) 2–9	**0.026**
	TMMY group	3 (4) 2–6	3 (4) 3–6	0.598
	**Walking forwards (steps), all**	15 (12) 10–15	15 (14) 14–15	0.107
	T group	13 (12) 7–15	15 (13) 8–15	0.655
	TPh group	14 (11) 4–15	15 (13) 13–15	0.068
	TMMY group	15 (15) 15–15	15 (15) 15–15	1.000
	**Walking backwards (steps), all**	15 (11) 5–15	15 (13) 10–15	**0.027**
	T group	9 (9) 2–15	10 (10) 4–15	0.180
	TPh group	15 (11) 4–15	15 (13) 9–15	0.109
	TMMY group	15 (14) 15–15	15 (14) 15–15	0.317
	**Weight (kg), all**	63.8 (66.7) 55.8–74.4	63.1 (67) 56.7–77.0	0.118
	T group	62.4 (68.8) 54.8–86.1	63.1 (69.4) 52.9–89	0.500
	TPh group	67.6 (68.6) 57.2–79.9	63.7 (68.5) 57.7–81.1	0.594
	TMMY group	59.4 (62.8) 52.7–70.4	58.9 (63.3) 53.6–70.6	0.176
	**Height (cm), all**	160 (161) 154–169	159 (161) 155–170	0.517
	T group	155 (158) 150–167	155 (157) 149–168	0.492
	TPh group	164 (164) 157–172	164 (163) 157–172	0.933
	TMMY group	156 (159) 154–165	157 (159) 154–165	0.581

Abbreviations: Md, median; and (X), mean. Bold text: *p* < 0.05. All *n* = 21; T group *n* = 5; TPh group *n* = 9; and TMMY *n* = 7.

**Table 5 ijerph-20-04933-t005:** Outcomes and the change between the baseline, post-intervention, and 1-year post-intervention follow-up.

		Baseline (1) Md (X) 25–75%	Post-Intervention (2) Md (X) 25–75%	1-Year Follow-Up (3) Md (X) 25–75%	*p*-Value (1) vs. (2)	*p*-Value (2) vs. (3)
**Fall**						
	**FES-I, all**	21 (26) 19–34	22 (26) 20–34	24 (26) 20–28	0.476	0.878
	T group	24 (30) 19–45	26 (30) 20–45	28 (33) 18–51	0.713	0.279
	TPh group	20 (23) 19–29	20 (22) 18–25	21 (22) 21–25	0.527	0.735
	TMMY group	24 (26) 18–35	31 (28) 20–35	27 (24) 19–29	0.236	0.207
**Physical activity**						
	**Physical exercise weekly (min), all**	30 (42) 8–75	45 (41) 0–75	45 (51) 0–98	0.524	0.086
	T group	30 (39) 30–53	45 (39) 0–75	45 (57) 23–98	0.890	0.336
	TPh group	15 (38) 0–75	38 (39) 0–68	30 (41) 0–90	1.000	0.705
	TMMY group	45 (49) 30–75	45 (43) 0–75	75 (58) 0–105	0.414	0.102
	**Everyday activity weekly (min), all**	225 (198) 98–300	225 (200) 98–300	225 (189) 83–263	0.863	0.552
	T group	120 (138) 60–225	225 (195) 75–300	225 (168) 45–263	0.066	0.102
	TPh group	300 (205) 45–300	225 (195) 75–300	225 (178) 38–263	0.498	0.593
	TMMY group	225 (231) 225–300	225 (210) 120–300	225 (216) 120–300	0.671	1.000
	**Total physical activity weekly (min), all**	255 (235) 116–323	255 (236) 128–334	255 (247) 146–345	0.948	0.628
	T group	150 (177) 90–278	270 (234) 75–375	225 (225) 90–360	0.144	0.684
	TPh group	263 (231) 49–375	232 (221) 101–326	255 (236) 79–371	0.610	0.750
	TMMY group	270 (281) 225–330	300 (253) 195–300	300 (274) 225–345	0.611	0.465
	**Daily sitting/resting (hours), all**	5 (5) 2–7	5 (5) 2–7	5 (5) 2–5	0.796	0.655
	T group	2 (4) 2–7	5 (4) 4–5	5 (5) 4–7	0.564	0.317
	TPh group	5 (5) 2–7	5 (5) 2–8	5 (4) 2–5	0.527	0.317
	TMMY group	5 (6) 5–8	5 (6) 5–8	5 (6) 5–8	1.000	1.000

Abbreviations: Md, median; (X), mean; FES-I, Falls Efficacy Scale–International. All *n* = 21; T group *n* = 5; TPh group *n* = 9; and TMMY *n* = 7.

## Data Availability

The datasets generated and/or analyzed in this study are not publicly available as the Ethical Review Board has not approved the public availability of these data.

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
