# Peer review of "Patient Education Improves Pain and Health-Related Quality of Life in Patients with Established Spinal Osteoporosis in Primary Care—A Pilot Study of Short- and Long-Term Effects"

_ijerph, 2023, doi:10.3390/ijerph20064933_

Round 1
Reviewer 1 Report
The study covers an important topic, that of providing appropriate education and training to patients affected by spinal osteoporosis and evaluates the impact of these interventions on several factors such as quality of life, symptom burden, physical activity and patient enablement.
While this is a pilot study and the sample size is small for making any generalizations, I believe it is very well constructed work with a very strong underlying methodology used. The first results are promising, and it will be very interesting to see what comes out of the final analysis.
I only have some minor comments.
-Firstly, it would be important to include some information on sample size calculation, and what is the enrollment goal for the completed study. Additionally, has the study protocol been registered?
- I believe there is a mistake on page 4, section 2.4 .Observation. While in Figure 1 the patients included in the baseline evaluation and allocation are 31, the text reports 21 patients analyzed. This should be better explained.
-It would also be interesting to have some information regarding which arm of intervention did participants lost at the 1 year follow-up belong to.
Reviewer 2 Report
Dear Authors,
congratulations for the concept of this article and the results of your work. Considering your study in a pilot phase there's no doubt that with a larger population your conclusions will achieve more consistency. Nonetheless there are some comments I would like to share:
- speaking about Osteoporosis implies a discrimination between primary (or primitive) and secondary, moreover a larger description of your populations (for example comorbidities, ADL, IADL) could broadly clarify if your multidisciplinary support has to be tailor made or standardized,
- the limitations occurred in this study are already described (number of patients, COVID lockdown, etc.), therefore I suppose that this pilot could be useful in order to re-design your study for a larger population,
- since we're talking about patients education, have you considered also to measure the level of education in your population?
- the instruments (questionnaire and tests) used totally suits with your purpose, therefore congratulations for choosing them.
Best Regards
